# Modulating the Antioxidant Response for Better Oxidative Stress-Inducing Therapies: How to Take Advantage of Two Sides of the Same Medal?

**DOI:** 10.3390/biomedicines10040823

**Published:** 2022-03-31

**Authors:** Priyanka Shaw, Naresh Kumar, Maxime Sahun, Evelien Smits, Annemie Bogaerts, Angela Privat-Maldonado

**Affiliations:** 1Research Group PLASMANT, Department of Chemistry, University of Antwerp, 2610 Wilrijk, Belgium; priyanka.shaw@uantwerpen.be (P.S.); maxime.sahun@uantwerpen.be (M.S.); annemie.bogaerts@uantwerpen.be (A.B.); 2Solid Tumor Immunology Group, Center for Oncological Research (CORE), Integrated Personalized and Precision Oncology Network (IPPON), University of Antwerp, 2610 Wilrijk, Belgium; evelien.smits@uza.be; 3Department of Medical Devices, National Institute of Pharmaceutical Education and Research, Guwahati 781125, India

**Keywords:** free radicals, reactive oxygen and nitrogen species, oxidative stress, antioxidants, human diseases, redox signaling

## Abstract

Oxidative stress-inducing therapies are characterized as a specific treatment that involves the production of reactive oxygen and nitrogen species (RONS) by external or internal sources. To protect cells against oxidative stress, cells have evolved a strong antioxidant defense system to either prevent RONS formation or scavenge them. The maintenance of the redox balance ensures signal transduction, development, cell proliferation, regulation of the mechanisms of cell death, among others. Oxidative stress can beneficially be used to treat several diseases such as neurodegenerative disorders, heart disease, cancer, and other diseases by regulating the antioxidant system. Understanding the mechanisms of various endogenous antioxidant systems can increase the therapeutic efficacy of oxidative stress-based therapies, leading to clinical success in medical treatment. This review deals with the recent novel findings of various cellular endogenous antioxidant responses behind oxidative stress, highlighting their implication in various human diseases, such as ulcers, skin pathologies, oncology, and viral infections such as SARS-CoV-2.

## 1. Introduction

The term “oxidative stress” was formulated by Helmut Sies [1] and refers to the imbalance between the production of oxidants and antioxidant defenses that can damage a biological system. Cells have a variety of defensive mechanisms to regulate the balance between the formation and elimination of reactive oxygen and nitrogen species (RONS) for regular cellular functioning. In(ex)tracellular RONS have both beneficial and harmful roles in the human body [2]. Low levels of RONS play an important role in supporting cellular life cycles, such as proliferation and homeostasis. On the other hand, high levels of RONS can result in oxidative damage to the cellular constituents (e.g., proteins, lipids, and DNA) and induce cell death. However, under certain circumstances, high levels of RONS are required to maintain health. For example, high oxidative stress is used during the oxidative burst to combat bacterial and fungal infections [3,4]. In the same way, therapies that generate high levels of RONS could be used to treat a variety of diseases, including neurodegenerative disorders [5], respiratory diseases [6], various types of cancers [7,8,9,10], among others. Some of the current therapies enhancing oxidative stress levels via exogenous physical sources of RONS include ionizing radiation, cold atmospheric plasma, photodynamic therapy, laser, and UV radiation [11,12]. However, the therapeutic use of some extracellular RONS therapies has been inadequate, largely due to insufficient knowledge about how antioxidants work to control diseases [13]. While some therapies aim to increase the RONS levels in cells, some others aim to reduce them, either by preventing their formation or enhancing their removal. Yet, ROS are essential in cell signaling, and this may be one of the reasons why some therapeutic antioxidant approaches produced unsuccessful results in vivo [14]. The main therapeutic benefits of pro-oxidant treatments can be expected from the generation of oxidants that cause (i) inhibition of endogenous antioxidants or (ii) activation of cellular activity.

It is essential to recognize how these antioxidant defense systems can be targeted for therapeutic use, considering the two faces of oxidative stress in health and disease. This review article presents the biologically relevant oxidants and their chemistry, the enzymatic systems and the redox signaling networks involved in response to oxidative stress, the therapeutic use of oxidative stress in health care, and the current external sources of reactive species for disease treatment. We discuss the relationship of pro-oxidant therapies with the antioxidant mechanisms in disease progression, including ulcers, skin pathologies, oncology, and viral infections such as SARS-CoV-2.

## 2. Biologically Relevant Oxidants and Their Chemistry

Under normal physiological conditions, the sequential reduction in oxygen through the addition of electrons leads to the formation of several reactive oxygen species (ROS) and reactive nitrogen species (RNS), which we refer to here as “oxidants” (Table 1). These oxidants energetically react with biomolecules in a non-selective way and can prompt the production of other oxidants. By evolution, the products of O_2_ reduction, the oxidants O_2_^−^ and H_2_O_2_, were harnessed by cellular systems for cell signaling as secondary messengers and housekeeping (phagocytosis) functions [3,15,16].

At first glance, RONS do not meet the criteria to be good signaling molecules: some oxidants are extremely reactive, which means that they are short-lived and, by default, removed once their signal is perceived. However, because oxidants are small, inorganic molecules, they can easily diffuse from their site of action to the targets. It is possible that their ability to transmit a signal could be mediated by the oxidation of target biomolecules. Yet, several questions remain unsolved, such as the source, concentration, and kinetics of oxidant formation. These all are important factors required to elucidate the physiological actions of oxidants and to better understand how oxidants transduce their cellular signals, as well as how they regulate the antioxidant response. We will now discuss the main biologically relevant oxidants.

### 2.1. (NO•)

Nitric oxide (NO•) is a small gaseous molecule with a well-known signaling function. It can freely diffuse through the cellular membrane within 1 s, and due to its hydrophobicity, it can translocate across the lipid bilayer without any barrier [17], in contrast to water. Hence, NO• can easily diffuse away from a NO• producing cell to another cell with lower NO• content. One of the well-studied physiological effects in which NO• plays a major role is the control of smooth muscle contraction and the flow of blood through the vessels. Furthermore, NO• suppresses platelet aggregation [18], mediates glutamate neurotoxicity [19], inhibits protein synthesis [20], lysis of cancer cells, cellular signaling, vasodilatation, immune response [21,22], and also has an antibacterial role [4].

Several enzymes can potentially produce NO•, but in animals, it is mainly produced by nitric oxide synthase (NOS). Stuehr and Nathan identified the generation of NO• by macrophages [23]. Activated macrophages are the major source of pathologically high levels of NO• usually found at sites of infection and inflammation, and the concentrations can be as high as 1 µM [24]. The expression of inducible NOS (iNOS) in macrophages allows the production of NO• at high rates as part of the non-specific immune response to infection [23,25]. However, the sustained production of NO• may result in damage to the host tissue since high concentrations of NO• can induce apoptosis [26]. The cytotoxicity threshold for NO• in human lymphoblastic cells was found to be 0.5 µM [27]. Damage by NO• has been associated with cancer, arthritis, and myositis [28,29]. It is known that NO• can directly modify the DNA and related repair enzymes [30,31]. Moreover, NO• can cause lipid peroxidation due to the depletion of antioxidants such as ascorbic and uric acid. The net effect of exposure to NO• depends on the rate of NO• production and its diffusion rate, the concentration of potential reactants, the level of NO•-producing enzymes (such as NOS), and the distance between NO•-generating cells and the target cells.

### 2.2. Superoxide Anion (O_2_^−^)

Superoxide anion (O_2_^−^) is one of the most biologically significant oxidants produced by our body. O_2_^−^ is short-lived, with a lifetime of 2 to 4 µs [32], whereas it is remarkably stable in many organic solvents (lifetime of more than 100 µs) [32]. McCord and Fridovich observed that superoxide dismutase (SOD) was an efficient scavenger of O_2_^−^ by catalyzing the conversion of O_2_^−^ into O_2_ and H_2_O_2_ [33]. After dismutation into H_2_O_2_ in blood cells, specifically in neutrophils, myeloperoxidase (MPO) is released into the phagosome, a vesicle formed around a particle engulfed by phagocytes (macrophages, neutrophils, and dendritic cells). The stimulation of neutrophils and macrophages increases the rate of production of O_2_^−^, as well as of NO•. MPO then catalyzes the formation of hypochlorous acid (HOCl), which is strongly microbicidal. HOCl is very reactive and causes oxidation and chlorination of biological molecules, as shown by the elimination of bacteria and fungi in vitro [34]. Alternative explanations for its antimicrobial activity include the buildup of high concentrations of H_2_O_2_, which might kill the pathogens directly or via the formation of OH•. The presence of SOD on the surface or in the periplasm of many bacteria is a well-recognized virulence factor and endows resistance to host phagocytes [35,36],
(1)$$O_{2}^{-}\overset{\mathrm{SOD}}{\to }H_{2}O_{2}\overset{{\mathrm{Fe}}^{2+}}{\to }2\mathrm{OH}•$$
(2)$$O_{2}^{-}+\mathrm{NO}•\to {\mathrm{ONOO}}^{-}\overset{H^{+}}{\to }\mathrm{OH}•+{\mathrm{NO}}_{2}•$$
which implies a direct role of O_2_^−^ in bacteria elimination. Moreover, two pathways by which O_2_^−^ results in cell cytotoxicity have been suggested. In the first one, O_2_^−^ is involved as a reducing agent for transition metal ions such as Fe^2+^: in this scheme, the reduced metal ion catalyzes the conversion of H_2_O_2_ to OH• (Equation (1)). The second one invokes the formation of ONOO^−^ from O_2_^−^ and NO• (Equation (2)).

Studies have shown that NO• may inhibit the first pathway, i.e., O_2_^−^-dependent lipid peroxidation, through the production of OH• in vitro [37]. In macrophages, the simultaneous formation of O_2_^−^ and NO• at least partly protect the cells from NO•-induced apoptosis [38]. In recent studies, it has been found that extracellular O_2_^−^ produced by LPS-stimulated macrophages induced Ca^2+^-mediated signaling and cell death in pulmonary endothelial cells [39]. This indicates that extracellular O_2_^−^ produced by other sources either crosses the cell plasma membrane or modifies cell surface proteins to mediate cell signaling. Despite this study, whether O_2_^−^ crosses the cell membrane to elicit a discrete intracellular signal remains controversial.

### 2.3. Hydrogen Peroxide (H_2_O_2_)

H_2_O_2_ is the least reactive oxygen species and remains stable under physiological pH and temperature in the absence of metal ions in vivo. The intracellular production of H_2_O_2_ has mainly a regulatory role, and the normal intracellular steady-state concentration of H_2_O_2_ is 10 nM or below [40]. The majority of H_2_O_2_ produced by mitochondria is initially originating from O_2_^−^, which is produced by mitochondrial enzyme complexes [41,42]. The H_2_O_2_ generation from mitochondria is in the range of 50 µmol kg^−1^ min^−1^ [43].

H_2_O_2_ is continuously produced in vivo [44] at physiological levels. H_2_O_2_ is a major component in redox signaling [44,45] and the major redox species operative in redox sensing, signaling, and redox regulation [46,47]. For such a function, a controlled transport of H_2_O_2_ across membranes is required. It has been shown that the permeability of the plasma membrane to H_2_O_2_ is a significant factor in cell susceptibility to extracellular H_2_O_2_ [48]. Extracellular H_2_O_2_ can diffuse into the cell, where it is rapidly decomposed by the intracellular antioxidants. However, the spatial distribution of H_2_O_2_ in cells and tissues is not uniform. There are substantial gradients resulting from the different concentrations of antioxidant enzymes in the cell compartments [44].

In addition, the integral membrane proteins that transport water, aquaporins (AQP), can also transport H_2_O_2_ within the cell [49], and they have been suggested as biomarkers for disease. AQP expression during stress conditions regulates the cell membrane integrity as well as cell communication [49]. AQP3 expression in human pancreatic cancer cell lines enhanced intracellular H_2_O_2_ levels in vitro (H_2_O_2_ = 80–90 µM) [48], highlighting its role in controlling the influx of H_2_O_2_ through the plasma membrane. In wound healing, H_2_O_2_ signaling has been established [48,50]. However, H_2_O_2_ may induce lipid peroxidation [7,51] at concentrations > 150 µM [52]. Furthermore, exogenous H_2_O_2_ can cause phosphorylation of tyrosine and activation of growth factors [53,54]. The pathological relevance of extracellular H_2_O_2_ is also well recognized in inflammation and injury responses [55].

### 2.4. Hydroxyl Radical (OH•)

Hydroxyl radical (OH•) has a very short half-life of about 10^−9^ s [56,57] and is produced from O_2_^−^ and H_2_O_2_ by the Haber–Weiss reaction [58] or by the breakdown of H_2_O_2_ through the Fenton reaction in the presence of metal ions [8,51,59]. Moreover, it can also be produced exogenously by multiple pathways, such as the decomposition of water due to ionizing radiation (radiotherapy), cold atmospheric plasma, or the photolytic decomposition of alkyl hydroperoxides [60]. OH• induces apoptosis through lipid peroxidation. This peroxidation occurs with a reaction constant of about k = 10^9^ M^−1^ s^−1^ [61]. The critical concentration of OH• required to induce apoptosis has not been established yet.

### 2.5. Peroxynitrite (ONOO^−^)

In 1990, the first papers suggesting that ONOO^−^ could be a biological oxidant and plays a role as a signaling molecule were published [37,62]. The lifetime of ONOO^−^ is 1 s at pH = 7.4 and T = 37 °C [63]. This is still long enough to allow ONOO^−^ to diffuse distances equal to cellular diameters. It was found to be likely that ONOO^−^ can cross the membrane of erythrocyte cells by two different mechanisms; in the anionic form through an anion channel, and in its protonated form (i.e., as ONOOH), by passive diffusion. Moreover, using model phospholipid vesicular systems, it was demonstrated that ONOO^−^ freely crosses phospholipid membranes. The rate of diffusion for ONOO^−^ crossing lipid bilayers was found to be k_D_ = 320 s^−1^, i.e., at least 30 times faster than the ONOO^−^ reaction with CO_2_ at normal conditions. Thus, the significance of ONOO^−^ as a biological effector molecule is determined not only by its reactivity but also by its diffusion rate. Since ONOO^−^ is relatively long-lived, it can reach critical in(ex)tracellular targets located at distant locations.

Much of the vascular and tissue injuries observed in certain models of inflammation are inhibited by either SOD or NOS inhibitors, suggesting that both O_2_^−^ and NO are important mediators of tissue injury [64]. ONOO^−^ is a strongly oxidizing compound, and its cytotoxicity depends on its ability to induce lipid peroxidation on polyunsaturated fatty acids (PUFAs) [65,66], resulting in lipid peroxidation chain reaction and reaching a plateau for (ONOO^−^) > 100–200 µM [67]. As a result, it affects the integrity of the lipid membrane, increases its permeability to drugs or other oxidants, and changes the membrane fluidity [7,59].

Large amounts of ONOO^−^ lead rapidly to necrotic cell death, whereas smaller amounts promote apoptosis [68,69]. ONOO^−^ rapidly inactivates glutathione peroxidase during apoptosis [70]. Cell death is induced by the ONOO^−^ generated through the spontaneous reaction of O_2_^−^ and NO•. Furthermore, the antibacterial effects of ONOO^−^ have also been documented. At physiological pH, 250 µM ONOO^−^ resulted in 50% mortality of *E. coli* [71].

## 3. Enzymatic Systems Related to Oxidative Stress

The dual action of reactive species in biological systems demands a fine control of the production and elimination of oxidants in the cells. The production of RONS by nitric oxide synthase and nicotinamide adenine dinucleotide phosphate (NADPH) oxidase is needed for normal physiological processes. Likewise, superoxide dismutase, catalase, and glutathione peroxidase are indispensable scavengers of RONS to preserve homeostasis. In this section, we describe the function of these enzymatic systems (Figure 1) in both normal and pathological conditions.

### 3.1. Nitric Oxide Synthase

In cells, NO• is produced by NOS by the conversion of L-arginine into L-citrulline [72]. Mainly, three types of NOS isoforms are identified: neuronal NOS (nNOS; type I NOS), inducible NOS (iNOS; type II NOS), and endothelial NOS (eNOS; type III NOS). The activities of nNOS and eNOS are calcium-dependent, whereas the activity of iNOS is fully activated at basal intracellular calcium concentration, so its activity is calcium-independent [73]. Of the three NOS isoforms, nNOS constitutes the predominant source of NO• in neurons and localizes to synaptic spines. RONS generated by iNOS in macrophages and smooth muscle cells causes oxidative-mediated cell death. In contrast, eNOS-generated RONS help maintain blood pressure and relaxation of blood vessels in the brain and heart [66]. Remarkably, it has been shown that during liver infection, a certain number of hepatic enzymes, including iNOS, localize to peroxisomes [74]. However, it was later found that only monomeric iNOS is found in peroxisomes in vitro, but it had lower activity than iNOS [75]. Although significant progress has been made to unravel the function of NOS over the years, it is still needed to address the role of NOS in cellular oxidative stress-mediated signaling in human pathologies.

### 3.2. NADPH Oxidases

Nicotinamide adenine dinucleotide phosphate (NADPH) oxidases, also called “NOX” enzymes, are constituted of several subfamilies of the membrane-bound protein complex. They help to transfer electrons across the plasma membrane to molecular oxygen, which results in the formation of O_2_^−^, H_2_O_2_, and OH● [76]. The NOX family contains different types of isoforms such as NOXs 1 to NOXs 5, DUOX1, and DUOX2. These isoforms generate a variety of ROS based on their type. For example, NOXs 1–3,5 produce O_2_^−^, whereas NOX4, DUOX 1, and DUOX 2 generate mainly H_2_O_2_ [77]. All the NOX isoforms participate in subcellular signaling processes [78]. The regulation and function of each NOX remain unclear, but it is believed that they are key mediators of normal biological responses. In addition, NOXs contribute to various diseases such as neuron inflammation, neurodegeneration, cardiovascular, renal disease, hypertension, and atherosclerosis [79]. Recently, several studies have demonstrated the involvement of the NOXs family in cancer [80] and neurodegenerative diseases. Hence, NOXs could be an interesting therapeutics target for the control of many diseases.

### 3.3. Superoxide Dismutase (SOD)

SOD plays a central role in ROS detoxification in the cell. SOD belongs to the group of metalloenzymes and catalyzes the dismutation of O_2_^−^ to H_2_O_2_ (Table 2) as well as molecular oxygen (O_2_), subsequently converting the potentially toxic effect of O_2_^−^ to a less hazardous compound. The enzyme SOD requires a metal cofactor to catalyze the dismutation of O_2_^−^ to O_2_ and H_2_O_2_ in a biological system. In humans, SOD binds to three metals: Cu/Zn (SOD1, cytosol), manganese (SOD2, mitochondria), and Cu/Zn (SOD3, extracellular) [81]. Cellular SOD concentrations are found virtually in all human tissues ranging between 4–10 µM. Since the amount of formed O_2_^−^ is controlled by SOD, this might be the reason for the cause of vascular and cardiovascular diseases upon SOD deficiency [82]. It is reported that mitochondrial superoxide dismutase-deficient mice present neurodegeneration and perinatal death [83]. Recently, Dayal and colleagues [84] showed that deficiency of SOD promoted vascular dysfunction and cerebral vascular hypertrophy in hyperhomocysteinemia. The deficiency of the SOD enzyme is quite common. Hence, the enzyme is essential to cellular health, protecting body cells from excess oxidants and other harmful agents that promote disease, specifically in cancer and aging. The levels of SODs decline with age, and as a result, the formation of free radicals increases. However, the clinical application of SOD as a therapeutic agent has been limited due to its extremely rapid plasma clearance time, instability, and immunogenicity in vivo. Several strategies have been proposed to overcome these problems but remain challenging.

### 3.4. Catalase (CAT)

Among all the antioxidant enzymes, CAT was the first enzyme to be discovered and characterized. It is present almost in all living tissues that use oxygen or manganese as a cofactor and catalyzes the dismutation of two molecules of H_2_O_2_ into water and oxygen. It is present in practically all types of living cells, where it scavenges H_2_O_2_ molecules (Table 2). CAT is primarily located in the peroxisome and helps maintain cellular homeostasis. It can break down thousands of H_2_O_2_ molecules within a second [85]. In particular, catalase is responsible for the clearance of exogenous H_2_O_2_ in vitro, and in intact, cultured human fibroblast cells, the rate of H_2_O_2_ removal was proportional to the cell density [86].

The mutation or deficiency of the CAT enzyme is associated with various diseases and abnormalities in humans. For example, polymorphism in the catalase-encoding gene resulted in oxidative DNA damage, the subsequent risk of cancer susceptibility, and the development of the mental disorder [87]. Additionally, low catalase levels in humans (acatalasemia) make them more susceptible to type 2 diabetes mellitus, atherosclerosis, and neoplasm [87]. Moreover, in cancer, modulating the catalase expression is emerging as a novel approach to potentiate chemotherapy. Recently, it has been shown that treatment designed to enhance cellular catalase reduces oxidative stress in rhinopathy and intestinal diseases [85]. Thus, targeting CAT may be a promising therapeutic approach to treat various diseases, including cancer and intestinal dysfunctions caused by the imbalance of the intracellular H_2_O_2_ level.

### 3.5. Glutathione Peroxidase (GPX)

Glutathione peroxidase (GPX) is a selenium-dependent enzyme present in the cytosol (GPX1 and GPX2), mitochondria (GPX4), and extracellular space (GPX3) [81,88]. It breaks down H_2_O_2_ to water, and lipid peroxides to their corresponding alcohol, depending on selenium-containing enzymes called selenocysteine peroxidases. GPXs play a crucial role in inhibiting the lipid peroxidation process by coupling its reduction to H_2_O with oxidation of reduced glutathione (GSH), a thiol-containing tripeptide (Glu-Cys-Gly), and therefore protecting cells from oxidative stress. Thus, GSH is required to complete the catalytic cycle. The product, oxidized glutathione (GSSG), consists of two GSH linked by a disulfide bond and can be converted back to GSH by glutathione reductase enzymes (Table 2).

GPX1 is abundant and present in almost all types of cells, whereas GPx2 and GPX3 are specifically located in the gastrointestinal tract and kidney, respectively. Commonly, the structure of GPX is tetrameric, but GPx4 is a monomeric form and differs in substrate specificity that breaks down phospholipid hydroperoxides. The enzyme also has a mitochondrial isoform that leads to oxidative stress-mediated apoptotic cell death [51,59]. Under normal conditions, it has been proposed that the extracellular GPX3 could protect cells from oxidative damage [89]. However, GPX3 has been involved in cancer as it could suppress tumor progression in cancer cells exposed to oxidative stress [90].

GSH is the most abundant thiol in mammals, and it is present in the cell at concentrations ranging from 1–10 mM [91]. The thiol-containing tripeptide in GSH helps maintain the defense against oxidative stress in tissues [92]. Moreover, GSH also protects the cells by reducing the radicals formed from antioxidants such as α-tocopherol and ascorbic acid [13,93]. GSH donates a pair of hydrogen atoms to oxidize GSH into GSSG, which is an indicator of the redox state. The decrease in GSH/GSSG ratio causes an overproduction of ROS that further leads to the reduction in GSH and other oxidants. Thus, changes in the GSH redox state can disturb the antioxidant machinery and cause damage to biomolecules. Among the GPXs, GPX1 and GPX4 are identified as therapeutic targets in various diseases, such as cardiovascular and cancer disease.

## 4. Redox Signaling Networks

Oxidative stress disturbs cellular ROS homeostasis via a redox relay mechanism [94], which leads to an increase in intracellular ROS levels and modulates key cell signaling pathways. Redox signaling is also crucial in regulating tumorigenesis, autoimmunity, neurodegenerative diseases, and loss of tissue regeneration with age [79,95,96]. Here we describe some of the key networks activated upon oxidative stress involved in health and disease.

### 4.1. Mitogen-Activated Protein Kinase (MAP Kinase) Pathway

MAP kinases are the key signaling molecules required in multiple cellular functions. Inadequate MAPK function can cause various types of cancer and inflammatory diseases. The three sub-pathways of MAPK include p38 mitogen-activated protein kinase (p38MAPK), extracellular signal-regulated kinase (ERK-1/2), and c-Jun-terminal kinase (JNK), which are the most widely studied among the many subfamilies of the MAPK family [97,98]. Under oxidative stress, the p38MAPK and JNK signaling pathways are mainly activated and play a role in inflammation, growth inhibition, and proapoptotic signaling (Figure 2) [99]. However, the ERK signaling pathway mainly participates in cell growth and development. However, the mutation of MAPK can lead to the development of diseases such as cardio-facio-cutaneous syndrome and kidney disease [100].

### 4.2. The Keap1-Nrf2-ARE Pathway

Kelch-like ECH-associated protein 1 (Keap1) is referred to as a negative regulator of the nuclear factor E2-related factor (Nrf2) [101]. It participates in cellular defense against various exogenous and endogenous stressors [59,101], and hence, it is a potential target for many drugs for the control of diseases. NRF2 belongs to the cap”-n”-collar subfamily of the basic-region leucine zipper bZIP transcription factors and binds to the *cis* element electrophile response element (EpRE) and other antioxidant response elements (AREs) in DNA. Together, they regulate the expression of more than 200 genes involved in antioxidant defense, DNA repair, proteasome activity, among other processes required for cell survival [14]. The system is induced by electrophiles such as H_2_O_2_ and other intermediary metabolites. In addition, Nrf2 inducers such as itaconate and tert-butylhydroquinone react with Keap1 (cysteine thiol groups), resulting in a defensive antioxidant response through the activation of the Keap1-Nrf2 signaling pathway (Figure 2) [102,103]. Upon oxidative stress, Nrf2 is released, eludes degradation, and translocates to the nucleus, where it starts its transcriptional activity on target genes. The dissociation of Nrf2 is mainly due to the conformational change in oxidized Keap1 (disulfide bond formation between Cys273 and Cys288). Multiple factors such as musculoaponeurotic fibrosarcoma and AREs in the Keap1-Nrf2 signaling pathway are responsible for the protection of cells under oxidative stress [104,105]. Any mutation in the Keap1 promoter or modification protein reduces its expression, resulting in a higher expression of Nrf2, which consequently favors cancer cell survival [106]. So, activation of Nrf2 might be a key target for cancer therapy [107,108]. Apart from cancer therapy, Nrf2 can also be a promising target for the development of neuroprotective and antidiabetic drugs [109,110].

### 4.3. Heme Oxygenase (HO)

Heme oxygenase (HO) is present in almost all types of mammalian tissues. It is an important rate-limiting enzyme that helps in the degradation of heme into various metabolites such as free ferrous iron (Fe^2+^), biliverdin, and carbon monoxide [111]. The byproducts are involved in various intracellular processes such as inflammation, oxidative stress, and cell death. There are two types of HO isoforms available, HO-1 (HMOX1 gene) and HO-2 (HMOX2 gene). Exposure to oxidative stress and hypoxic conditions destabilizes HO and induces HO-1 [112], which may further protect the cells against oxidative stress. [113]. Regardless of its role in heme catabolism, HO-1 participates in various diseases such as immunomodulation and skin diseases (vitiligo and psoriasis). Thus, targeting the HO-1 could be the remedy to treat such diseases.

### 4.4. NF-κB Pathway

The transcription factor nuclear factor κB (NF-κB) is localized in the cytoplasm as a heterodimer. Once activated, NF-κB translocates into the nucleus to induce gene transcription of genes related to cell survival. The activation of NF-κB occurs at the early stages of oxidative stress, but it is reduced under the sustained presence of ROS [114]. ROS activates the canonical pathway by inhibiting the phosphorylation of I*κ*Ba (required to block the binding of NF-κB to the DNA) and inhibiting IKKβ activity [54].

The NF-κB activity influences the levels of ROS in the cell by increasing the expression of antioxidant proteins such as CAT, SOD, GPX, ferritin heavy chain, thioredoxins, and glutathione S-transferase pi, HO-1, among others. Conversely, ROS can repress NF-κB signaling, depending on the phase and context. The direct oxidation of NF-κB on the cysteine residue Cys-62 in the RHD domain of the p50 subunit inhibits NF-κB binding to the DNA [115]. In addition, the NF-κB pathway regulates the expression of pro-oxidant targets such as iNOS [116], cytochrome p450 enzymes [117], xanthine oxidase/dehydrogenase [115,117,118], and NADPH oxidase NOX2 subunit gp91phox [114], among others.

## 5. Application of Oxidative Stress in Health Care

### 5.1. Wound Healing

The role of antioxidants is critical in wound and tissue regeneration, as maintaining the redox balance is crucial to regulating the different phases of the healing process. At low concentrations, ROS help recruit immune-competent cells, promote cell proliferation, and favor wound repair. However, defective ROS detoxification can cause senescence and apoptosis, bacterial colonization, and chronic inflammation [3]. As a result, chronic wounds (vascular, diabetic, and pressure ulcers) with a highly oxidative environment can develop (Figure 2) [119]. The most common treatments involve antioxidant strategies that target the mitochondria, as it is where ROS are produced. The impaired oxygenation due to vascular disruption and depletion of O_2_ causes hypoxia, which favors ROS production and reduces the antioxidant defenses [3].

Currently, there is a range of studies on ointments and hydrogel dressings combined with antioxidant compounds that could reduce the oxidative stress in the wound, as recently reviewed [120,121]. The biomaterial chosen must have excellent physicomechanical and chemical properties that support cell growth and wound repair [122]. The ant-oxidants used in combination with biomaterials can be different: thiol compounds (NAC, GSH, γ-glutamyl-cysteinyl-glycine), non-thiol compounds (polyphenols such as curcumin and anthocyanins) [123,124], vitamins (such as ascorbic acid, α-tocopherol, vitamin A, C, and E), and antioxidant enzymes (catalase, GSH-reductase, GSH-peroxidase) [121]. While many antioxidants can be bought over the counter or by prescription, only one has been approved by the FDA for wound healing. This is the case of medical honey-like Medihoney, a medical-grade product for the treatment of wounds and burns, containing glucose oxidase and *Letospermum* compounds with antibacterial properties [125]. Glucose oxidase breaks down glucose into gluconic acid and H_2_O_2_, the latter being responsible for the antibacterial activity of honey [126]. However, the free radicals produced by H_2_O_2_ are neutralized by the flavonoids and other polyphenols present. The low pH of honey (infected and recalcitrant wounds present higher pH values), the stimulation of angiogenesis, granulation, and epithelialization contribute to the healing process [125]. In addition, a new range of wound dressings with intrinsic antioxidant properties that do not require the addition of antioxidant compounds is currently under study.

A contrasting approach aiming to deliver or generate RONS to wounds has also proved to enhance the healing process. The use of CAP in patients with acute and chronic wounds has been shown to reduce ulcer size, accelerating the healing process when used alone or in combination with standard wound care procedures [127,128,129,130]. While CAP relies on the delivery of RONS, this is not comparable to radiotherapy, as CAP-derived RONS are delivered to the tissue in a localized and controlled manner. CAP treatment has been shown to improve wound oxygenation, cell migration for tissue repair and has antimicrobial activity [131]. The application of CAP on keratinocytes under chronic redox stress in vitro promoted the antioxidant phase II response elements such as GPX1, GPX5, and GPX8 at the beginning of the chronic oxidative challenge, and SOD 1/3, peroxiredoxin PRDX2 at later stages [130,132]. These findings suggest that mild CAP treatment might accelerate wound healing by modulating the redox signaling pathways (Nrf2-ARE targets HMOX-1, GSR, NQO1, SOD, GSH), regulating cell communication via cytokines/chemokines, and finally promoting cell migration into the wound bed [133].

The decision to use pro- and antioxidant treatments for wound care is complex, and the selection of treatment should be made depending on the stage of the wound healing process, the type of wound (exudating or desiccated), the level of inflammation and microbial infection, and the presence of necrotic tissue.

### 5.2. Skin Pathologies

In the skin, multiple biochemical processes take place, including the generation of ROS (Figure 2). A common feature of many skin disorders is the reduced levels of antioxidants, both enzymatic and non-enzymatic. In some skin pathologies, there is clear evidence of oxidative stress. Melasma patients present symmetrical hyperpigmentation caused by the overproduction of melanin. These patients have high levels of malonaldehyde (MDA, a critical biomarker for lipid peroxidation), NOS, SOD, and GPX in serum [134,135], which suggest an active state of oxidative stress. The current therapies include ointments containing hydroquinone to reduce melanogenesis, corticosteroids to reduce inflammation, and tretinoin, a retinoid with antioxidant properties that improve keratinocyte turnover [136].

In contrast, patients with vitiligo have a reduced number of melanocytes and, therefore, reduced production of melanin. It has been found that melanocytes from patients with active vitiligo are exposed to abnormal levels of oxidative stress, which could be a consequence of abnormal mitochondrial ROS production, low production of antioxidants, and expression of pro-apoptotic proteins such as TRAIL [137,138]. Here, catalase expression is reduced, and the protein is inactivated, leading to the accumulation of high levels of H_2_O_2_ and the destruction of melanocytes [139]. Clinical studies using pseudo catalase creams alone or in combination with SOD to mitigate the oxidative damage caused in melanocytes have been unsuccessful [140]. However, the combination of pseudo catalase, calcium, and UVB light therapy has shown promising results, although more studies are needed to confirm these findings [141,142]. Unfortunately, no antioxidant therapies are currently approved for the treatment of vitiligo.

In alopecia, patients show a disrupted SOD response, reduced levels of GPX [82], HO-1 [111,112,113,143,144], thioredoxin reductase [145], β-carotene, and vitamin E [146], but unchanged catalase expression [147]. It has been suggested that the hair follicle cell apoptosis induced by oxidative stress is a possible cause of alopecia [148]. Whereas it is acknowledged that ROS plays a significant role in this condition [149], there are currently no approved therapies to reduce oxidative damage.

Dermatitis is another skin condition that can present in a variety of forms. In atopic dermatitis, infiltrated immune cells release proinflammatory cytokines and ROS, and it has been suggested that the pathophysiology of atopic dermatitis could be related to the impaired antioxidant response [150]. Patients often present reduced levels of SOD, catalase, glutathione peroxidase, GSH, vitamin A, E, and C [151]. Current therapies include the application of emollients, steroids, calcineurin inhibitors, and phototherapy [152], although some patients opt for the use of dietary supplements such as vitamin D or natural products. To date, there are no approved antioxidant treatments that can help control the condition, alone or in combination with other compounds.

In the chronic inflammatory autoimmune disease lichen planus (LP), it has been found that patients with oral LP present increased SOD and reduced glutathione peroxidase levels in saliva [153]. In addition, patients with cutaneous LP showed low levels of serum and tissue catalase, reduced activity of GPX, and elevated serum levels of other pro-oxidants [154,155]. This disbalance creates an excessive amount of H2O2 that accumulates in cells and causes damage. Although the most common therapies include steroids and calcineurin inhibitors, retinoids are also administered as they have immunomodulatory and antioxidant properties [156].

In psoriasis, there is abnormal metabolism of fatty acids and ROS generation. The high levels of MDA suggest an increase in the peroxidation of the cell membrane. The reduced levels of plasma beta-carotene and α-tocopherol, together with the reduced antioxidant enzymatic activity of catalase and glutathione peroxidase, exacerbate the disease [150]. The current therapy for psoriasis includes retinoids with antioxidant properties such as acitretin, tazarotene, and calcipotriene. Conversely, anthralin (a hydroxyanthrone that accumulates in the mitochondria, increases ROS production, and blocks DNA synthesis) is also approved for the treatment of psoriasis [157].

The literature suggests that antioxidants are unable to singlehandedly cure these skin pathologies, as their etiology might involve other factors such as defective immune responses or stress conditions other than oxidative damage. However, the role of ROS in the pathogenesis of these diseases is acknowledged, and further studies could bring light to novel therapeutic approaches to modulate the antioxidant response.

### 5.3. Oncology

Cancer cells produce higher levels of RONS than normal cells, which alter pro-oncogenic signaling pathways that favor the prevalence of a malignant phenotype. For cancer cells to survive this high oxidant environment, they need an effective antioxidant response. High levels of antioxidants are required for the start, progression, and metastasis of different types of cancers (Figure 2) [158]. Several transcription factors are involved in the control of this response. The main regulator of antioxidant genes is the Nrf2 transcription factor that drives cancer progression, invasion, and metastasis. Nrf2 is involved in each of the hallmarks of cancer directly (by upregulating its target genes) or indirectly (by modulating the redox state) [102,103,106,159,160,161]. Upon induction of oxidative stress, Nrf2 translocates to the nucleus, where it binds to genes containing AREs. This way, Nrf2 promotes the expression of antioxidants such as NADPH quinone oxidoreductase 1, heme oxygenase-1, ferritin heavy polypeptide 1, and the cystine/glutamate antiporter SCL7A11 [162]. Oncogenes such as K-RAS, BRAF, and c-MYC stabilize Nrf2 [163], which can promote chemotherapeutic resistance, as observed in pancreatic, colorectal, and ovarian cancers [164]. The NF-κB transcription factor plays a protective role in cancer by suppressing the accumulation of toxic ROS, increasing the MnSOD and thioredoxin levels in cells, and upregulating antiapoptotic genes [164,165]. NF-κB modulates autophagy in cancer cells, and at the same time, autophagy can modulate NF-κB signaling [166]. Interestingly, there is a complex interplay between Nrf2 and NF-κB pathways as they modulate each other.

The antioxidant response mounted by cancer cells also participates in metastasis, and the survival of cancer cells once detached from the extracellular matrix (ECM). Under normal circumstances, ECM detachment results in the induction of anoikis, a caspase-dependent cell death mechanism that involves a steep increase in ROS production. However, cancer cells can prevent anoikis by fortifying their ROS defenses: they maintain NADPH production and inhibit its consumption, synthesize GSH, and use alternative antioxidant pathways to compensate for any blocked pathways [167]. This suggests that antioxidants are critical for the survival of ECM-detached cells.

Cytoglobin is a protein present in all cells that not only can transport oxygen and scavenge RONS but also can suppress tumor growth. It has been shown that in some types of cancer, cytoglobin becomes hypermethylated, which silences its expression and promotes tumor progression [168,169]. Conversely, the overexpression of cytoglobin in head and neck cancer patients seems to correlate with increased aggressiveness of the disease, which could be linked to the increased hypoxic state of the tumor [170]. As a RONS scavenger, the heme group in cytoglobin becomes nitrated upon binding to NO2• due to its NO dioxygenase activity [171]. In addition, cytoglobin has peroxidation activity, which means that it can consume both hydrogen and lipid peroxides. At low concentrations of cytoglobin in an oxidative environment, the binding of one lipid unit per cytoglobin allows the oxidation of the lipid, producing vasoactive isoprostanes or electrophilic lipids that can affect multiple cell signaling pathways [172]. These cell signaling molecules then allow cells to respond promptly to the stress by either boosting their antioxidant response to prevent further damage or inducing apoptosis.

The most well-known cancer therapies that increase ROS levels beyond the threshold are ionizing radiation and chemotherapy. After ionizing radiation, persistent oxidative stress characterized by mitochondrial dysfunction and upregulation of the NADPH oxidase complex is observed [173]. Likewise, chemotherapeutic drugs such as platinum-based drugs, adriamycin, and cyclophosphamide, impair the normal mitochondrial function and antioxidant response that increases superoxide anion [174].

Cancer patients that receive chemo- and radiotherapy have a reduced antioxidant status, as their antioxidant defenses are quickly depleted after treatment. It is estimated that between 13% and 87% of oncological patients use antioxidant supplements [175] in an attempt to protect the healthy cells from the toxicity of radio- and chemotherapy and prevent cardiac damage, pulmonary complications, and fertility problems [176]. However, the relationship between chemo- and radiotherapy, on the one hand, and the antioxidant systems, on the other hand, is complex. It has been suggested that patients with a low antioxidant status may present a higher neoplastic activity and poor health and could benefit from antioxidant supplementation to improve their survival and quality of life [177]. Nevertheless, there are concerns about the possible negative effect of antioxidant supplements in patients undergoing therapy. While preclinical studies have shown positive responses in animals that received antioxidants together with chemotherapeutic drugs, these findings were challenged when the studies were translated to humans with various types of cancers [178]. Some of the antioxidants challenged include vitamin C, vitamin E, GSH, and β-carotene, among others. However, mixed outcomes are reported in the literature due to the lack of sufficient and well-designed clinical trials, making it difficult to consolidate these findings for therapeutic use. While antioxidants could be administered before or after therapy, their administration during radio- or chemotherapy is not recommended, as this could protect the tumor and reduce the survival of the patient [179].

### 5.4. Respiratory Viral Infections

Human respiratory viral infections constitute a group of diseases that affect millions of people worldwide, especially kids, immunocompromised and elderly people, leading to substantial morbidity, mortality, and economic losses worldwide, as seen in the current COVID-19 pandemic [180]. Respiratory viruses (influenza, human respiratory syncytial, human rhinovirus, human metapneumovirus, parainfluenza, adenovirus, coronavirus) can infect the upper and/or lower respiratory tract in humans, causing common clinical signs and symptoms such as sore throat, nasal congestion, cough, and fever, or more specific and severe manifestations, such as pneumonia, bronchiolitis, and severe acute respiratory syndrome [181].

Viruses are obligated intracellular parasites that hijack host cellular machinery to replicate. Viral infections, therefore, induce an imbalance in the intracellular microenvironment, affecting, among other systems, the redox system [182]. Generally, respiratory viruses induce ROS-generating enzymes, such as NADPH oxidase, NOX, and xanthine oxidase (XO), and lead to an increase in the production of RONS (e.g., OH•, O_2_^−^, ONOO^−^, HClO, H_2_O_2_, and NO•) and a depletion of antioxidants (e.g., NADPH, SOD, CAT, and GPX). RONS plays an ambiguous role in respiratory viral infections depending on their production, cell type, and virus involved. On the one hand, RONS are seen as a protection mechanism for the host cell against pathogens. As central components of the “respiratory burst”, RONS initially fight infection by activating leukocytes, which might contribute to the induction of apoptosis. On the other hand, the oxidative stress caused by the viral infection can contribute to several aspects of pathogenesis, including inflammatory responses, cell death, weight loss, tissue damage, cell-to-cell viral transmission, and robust cytokine and chemokine production, leading to cytokine storms. Even when little is known about the mechanisms involved in the imbalance of the redox systems caused by a virus infection, the use of antioxidants as therapeutics or the modulation of RONS and oxidative stress could represent an interesting but challenging pharmacological approach in the battle against respiratory viruses [182,183,184]. In the context of the COVID-19 pandemic, this approach is currently investigated to better understand SARS-CoV-2 pathogenesis and identify possible therapeutic targets.

### 5.5. Oxidative Stress and Antioxidants in SARS-CoV-2 and Potential Therapeutics

The novel coronavirus known as severe acute respiratory syndrome coronavirus 2 (SARS-CoV-2) is a positive-sense, single-strand enveloped RNA virus capable of infecting mammalian and causing respiratory, gastrointestinal, and central nervous system diseases (Figure 2) [185,186,187]. Around 15% of COVID-19 patients suffer from severe pneumonia and 5% from organ failure, toxic shock syndrome, or acute respiratory distress syndrome (ARDS) [188,189]. Many studies suggested cytokine storm as a principal factor in the development of ARDS [190], and the link between proinflammatory cytokine signaling and oxidative stress is actively investigated in the context of COVID-19 infection. The excessive production of RONS during oxidative burst following SARS-CoV-2 infection could be a possible mechanism inducing severe lung pathology [191,192]. Importantly, the deteriorating effect of RONS not only affects the respiratory epithelium but also other cell types, such as erythrocytes, which could be linked with hypoxic respiratory failure observed in some patients with COVID-19 [193,194]. The possible resulting free hemoglobin and heme concentration increase could, in turn, enhance oxidative stress. Moreover, the erythrocytes membrane is altered by amplified RONS generation leading to phagocytosis in macrophages and neutrophils [190]. Recent clinical studies investigating the oxidants-antioxidants balance in COVID-19 patients highlighted the role of oxidative stress in the infection. Firstly, the levels of antioxidant vitamins (A, C, E), enzymes (glutathione, SOD, catalase), and trace elements (manganese, zinc, selenium, etc.) were found to be reduced, suggesting alteration of the redox state. Secondly, the severity of COVID-19 for elderly patients seems to be associated with the downregulation of some redox-active genes (SOD3, ATF4, M2TA) observed in the lungs. Finally, in parallel to the decrease in the levels of antioxidants, an increase in both oxidative stress and levels of RONS was observed in the severe forms of the disease. Different cellular and molecular pathways have been proposed to further explore the link between SARS-CoV-2 infection and increased oxidative stress. More precisely, the downregulation of ACE2 expression on the SARS-CoV-2 infected endothelial cell surface leads to endothelial dysfunction and vascular inflammation-inducing the imbalance of the renin-angiotensin-aldosterone system and triggering the production of RONS via NOX activation and reduced availability of NO• via decreased eNOS activity [195].

As previously described, viruses such as SARS-CoV-2 highjack host cell machinery and establish favorable conditions for viral replication via the increase in oxidative stress caused by an excess of RONS and a deficiency of antioxidants [196]. This oxidative environment favors the binding of the SARS-CoV-2 spike protein to ACE2 [197]. Thereby, targeting oxidative stress by modulating the sensitive redox pathways to regulate the immune response represents a promising therapeutic approach to fighting SARS-CoV-2 infection. Among several natural products, vitamins, and compounds with anti-inflammatory and antioxidant properties already tested, we can mention NAC, GSH, polyphenols, vitamins C, D, and E, melatonin, pentoxifylline, selenium, high-dose zinc, or inhaled nitric oxide [182,195,198,199,200]. To date, even though data have been collected, the efficacy of such treatments targeting oxidative stress is still controversial, and more research is necessary [195,201]. For example, a clinical trial revealed that treatment with antioxidant supplements (vitamin C and E, N-acetylcysteine, melatonin, and pentoxifylline) was found to reduce the severity and lethal outcomes of COVID-19 infection [185,187,193,201]. Likewise, pulmonary circulation of COVID-19 patients with severe pneumonia was improved after nitric oxide inhalation. Nevertheless, other clinical trials do not report a significant reduction in symptom duration, days of hospitalization, the proportion of patients requiring intubation, or overall mortality after antioxidant supplementation.

## 6. Source of Oxidants and Free Radicals in the Treatment of Disease

Oxidants can be derived from endogenous and exogenous sources (Table 3). These oxidants are continuously formed in the cells as a consequence of both enzymatic and non-enzymatic reactions. Endogenous free radicals formed during immune cell activation, inflammation, stress, excessive exercise, ischemia, infection, cancer, and aging are mainly produced via enzymatic reactions. Exogenous RONS can originate from industrial solvents, radiation, as well as from certain drugs used for medical treatments (cyclosporine, tacrolimus, gentamycin, bleomycin), etc. After penetration into the body by different routes, these exogenous compounds alter the normal redox status, which leads to disturbance of the cell signaling, further leading to activation or deactivation of signaling pathways [202,203]. There is a connection between the levels of oxidants in a cell and the activation of MAPK signaling. Especially, MAPKs are activated by H_2_O_2_ [204,205] led to the general recognition that RONS-signaling pathways have an important function in cell proliferation and growth (via extracellular signal-regulated kinase (ERK) mitogen-activated protein kinase (MAPK pathway) [95,206], and transcription factors. However, high levels of intracellular RONS, induced by the extracellular sources, are found to be controlling cancer by regulating the genes that are involved in metabolism, metastasis, and angiogenesis. There is also an intricate relationship between RONS and the immune system [207,208].

Several physical and chemical modalities are used as exogenous delivery of RONS to treat human diseases via the regulation of redox signaling. They will be described in the following subsections.

### 6.1. Physical Sources

#### 6.1.1. Radiotherapy

Radiation therapy uses high-energy particles or waves, such as x-rays, gamma rays, electron beams, or protons, that generate RONS [209,210] (Table 3) and induce intracellular oxidative stress in the subcellular compartments [211,212]. The standard therapy involves the application of a total dose of 40–50 Gy for the treatment of breast cancer [213] and rectal cancer [214]. However, the recommended doses sometimes fail to ablate the tumor and can lead to resistance due to the inherent characteristics of the tumor [215]. In addition, the presence of cancer stem cells (CSCs) in the tumor can contribute to the resistance: CSCs have low intracellular ROS levels, an increased expression of ROS scavengers, an efficient DNA repair system, and can inhibit apoptosis [216].

Besides cancer therapy, low doses of radiotherapy can be used to treat non-neoplastic degenerative, chronic inflammatory, or proliferative diseases, as it blocks different inflammatory mediators and promotes the production of anti-inflammatory cytokines [217]. For example, the treatment with <1 Gy can inhibit iNOS expression and reduce endothelial cell-leukocyte interactions due to a decrease in the expression of adhesion molecules and reduced vasodilatation [218].

#### 6.1.2. Photodynamic Therapy (PDT)

PDT is a non-invasive and efficient strategy based on photo-physical principles that may provide specific oxidative damage in organelles such as the endoplasmic reticulum, mitochondria, and lysosomes [219] (Table 3). In PDT, the photosensitizing chemical substance is activated by light in conjunction with molecular oxygen. PDT is widely used in treating acne, wound healing, and malignant cancers, including head and neck, lung, and skin cancer, but also for atopic dermatitis, vitiligo, and rare diseases such as mycosis fungoid (a type of cutaneous T-cell lymphoma) and sclerotic skin disease [219,220,221]. The activated photosensitizers transfer energy to O_2_, generating RONS [222]. The oxidant formed upon irradiation, mainly singlet oxygen (^1^O_2_), has a limited lifetime and ability to migrate from the site of formation. Thus, it interacts with biologic substrates only in the site where the photosensitizer was applied and triggers oxidative stress-mediated pathways (endoplasmic reticulum stress) [220] that help manage many cutaneous inflammatory dermatoses, as well as cancer.

Another strategy is to use pH-activated agents encapsulated in liposomes or polymeric micelles for cancer treatment. These agents respond to the high acidity of the tumor microenvironment and glutathione-bonded photosensitizer since the glutathione concentration is also higher in cancer cells [223]. The ability to combine PDT with potent biological agents and its cost-effectiveness makes PDT the preferred treatment for difficult-to-manage diseases. However, the safety of PDT in patients is still of concern. The most common acute side effect of PDT reported is a red phototoxic reaction that occurs about 24 h after treatment and causes severe pain [221].

#### 6.1.3. Laser Therapy

Nowadays, the laser is one of the most popular therapies in the skincare industry, and it can treat both hard and soft tissues. Specifically, low-level laser therapy (LLLT) operating at wavelengths of 600~1000 nm is used for analgesia, helps in tissue regeneration, and decreases inflammation by activating a variety of growth factors, such as vascular endothelial growth factor (VEGF), and transforming growth factor (TGF)-α and −β [224]. Moreover, laser treatments are also used in different medical fields such as dental care [225,226] and gingivectomy procedures [225,227] by promoting the re-epithelialization of cells at a faster rate [228,229]. Like PDT, laser therapy also works through photoreceptor systems in the mitochondria, which further leads to the generation of oxidants (especially ROS) (Table 3) [230]. At low levels, it activates tissue repair processes [231] and promotes the secretion of growth factors [224]. In contrast, high levels of exposure can lead to lipid and protein damage. Depending on the need, it is possible to control the exposure time and dose of the laser treatment [225,232], which further controls the excessive production of oxidants. Additionally, several studies have shown that laser therapy inhibits tumor necrosis factor-alpha (TNF-α), cyclooxygenase-2, interleukin (IL)-1β, and prostaglandin E2, which serves an important role in the induction of proliferation and survival of cancer cells [224]. However, these studies have provided conflicting results on the efficacy of laser therapy, which highlights the need for further studies.

#### 6.1.4. Cold Atmospheric Plasma (CAP)

Recently, CAP has emerged as a new therapy that can deliver RONS for biomedical applications [4,7,51,59,225,233,234,235]. CAP is a multi-component, chemically active, and highly reactive ionized gas that is generated at room temperature under atmospheric conditions, usually from noble gases (i.e., helium or argon), and flows into ambient air or is directly created in air. The species created by CAP are mainly RNS, such as NO• and nitrogen dioxide (NO_2_), as well as ROS, such as ozone (O_3_), OH•, O_2_^−^, ^1^O_2_, and H_2_O_2_ (Table 3) [7,9,59,236]. These RONS are formed in significant amounts by CAP devices, where the concentration of H_2_O_2_ increases with increasing humidity in the feed gas due to an increase in the OH• density. In contrast, the RNS concentration is unaffected by changes in environmental humidity. It has been reported that CAP-derived RONS (especially OH•) can be transported to millimeter depths to reach deep-seated diseased cells [237]. CAP has shown to induce a variety of biological effects, such as blood coagulation [238], tissue regeneration [237,239], sterilization [4,238,240], wound healing [15,228,238], cancer cell death [7,9,51,59,96,233,238,240,241], activation of immune cells [242,243], and virus inactivation [244,245]. The type and concentration of CAP-generated species delivered to cells depend on the CAP operating conditions, controlled by the design of the source, including the configuration of the electrodes. As CAP can activate the immune cells [246,247], it may also be beneficial in combination with immunotherapy for cancer treatment.

#### 6.1.5. Oxidant-Rich Liquids

The direct application of CAP to human tissues is approved by the European Committee for Standardization (CEN) and the International Organization for Standardization (ISO) for wound healing and head and neck cancer [248]. However, the direct application of CAP has some limitations due to the limited feasibility of delivering RONS to internal target tissues. It has been shown that it is also possible to use the CAP-derived RONS more flexibly by treating physiological solutions used in the clinic with CAP and administering these as treatment [4,7,51,59,234]. In this process, the CAP-derived RONS are delivered from the plasma gas phase into the liquid phase, yet leaving a delicate mixture of long-lived RONS (Table 3), able to further recombine or react again to form intracellular short-lived species [51,248]. In the future, these liquids could be used in the clinic. Some of the solutions used to generate plasma-treated liquids (PTL) include water [59], culture media [247], Ringer’s lactate [249], phosphate buffer saline (PBS) solution [250], Ringer’s solution, and bicarbonate Ringer’s solution [250]. Long-lived species of PTL components (namely, H_2_O_2_ and NO_2_^−^) show strong synergy with tumor suppressor enzymes, which are located on the cell membrane. Moreover, ONOO^−^ is produced from H_2_O_2_ and NO_2_^−^, followed by the primary ^1^O_2_. This ^1^O_2_ causes inactivation of membrane-associated catalase [51,251]. Other reactive species can also be derived from the solutes of PTL-exposed solutions [252]. For example, an NMR analysis showed that acetyl- and pyruvic acid-like groups are generated in Ringe’s lactate solution treated by CAP, which has shown a crucial antitumor role [253]. However, to effectively use PTL in the clinic, a clear understanding of the interaction of RONS with biomolecules (lipids, proteins, and nucleic acids) from the atomic to the macro scale, and their biological significance, is needed.

### 6.2. Chemical Sources

Chemicals (Table 3) such as cisplatin, temozolomide, carboplatin, doxorubicin, etc., induce oxidative stress-mediated lipid peroxidation in cancer [51,254,255]. These compounds induce cancer cell death by releasing electrons from the electron transport system (ETS) to O_2_ by NADPH dehydrogenase, resulting in the formation of O_2_^−^.

The oxidative stress caused by chemotherapeutic drugs interferes with cellular processes by changing the integrity of the cell membrane to cause cytotoxicity and thereby increasing their cytotoxic effect [7,51,243]. In addition, since some side effects caused by chemical agents appear to be prevented by certain antioxidants, administering these supplements during chemotherapy may diminish the development of side effects, as well as improve the response to therapy [256].

**Table 3 biomedicines-10-00823-t003:** Exogenous and endogenous sources of oxidants and their use in the treatment of various diseases.

Source	Therapy	Oxidants	Medical Conditions
Physical	Radiotherapy	Mainly OH• radicals	Cancer [93,94], meningiomas and neurinomas, prevention of cardiovascular restenosis [95]
	Photodynamic therapy (PDT)	ROS, H_2_O_2_, and ozone (O_3_)	Acne, wound healing, and malignant cancers, including head and neck, lung, and skin cancer [100,101,102]
	Laser therapy	ROS activation	Skin treatments (acne, rosacea, eczema), tissue repair, and mitochondria photostimulation [105,106,107,108,109,110,111,112,113,114]
	Cold atmospheric plasma (CAP)	NO•, NO_2_, O_3_, OH•, O_2_^−^, ^1^O_2_, H_2_O_2_, ONOO^−^ etc.	Cancer [12,14,15,16,18,23,118,119,120], wound sterilization [4,23,118], wound healing [23,24,109], tooth bleaching, beautification of the skin [116,117], and inactivation of viral infection [123,124]
	Oxidant-rich liquids; plasma-treated liquids (PTL)	Mainly long-lived species (H_2_O_2_, NO_2_^−^, ONOO^−^)	Cancer cell death (apoptosis, necrosis, and ferroptosis) [4,5,12,15,18], immunogenic cell death [132], sterilization (removal of biofilm), wound healing [238]
Chemical	Cisplatin, temozolomide, dozorubicin, doxorubicin, epirubicin, daunorubicin, carboplatin, and oxaliplatin, etoposide, teniposide, topotecan, irinoteca, etc.	ROS	Cancer and immunogenic cell death [12,15,122,133,134,135]
Intracellular components	Activated macrophages	NO•	Neurodegenerative diseases [4,257], tissue regeneration [21,25]
	Nitric oxide synthase (NOS)		
	NADPH oxidases (NOX)	O_2_^−^, H_2_O_2_, and OH•	Neurodegenerative and cardiovascular disease [5,76,79,257]

## 7. Conclusions

The generation of reactive species is an important and evolutionarily conserved bioprocess that can activate discrete signaling transduction pathways or disrupt redox cellular homeostasis, depending on their concentration. However, oxidative stress will occur when excessive reactive oxygen and nitrogen species (RONS) generation is induced, either by exogenous or endogenous sources, which will trigger many physiological and pathophysiological processes, such as autophagy, apoptosis, and necrosis. By targeting this homeostasis condition, we can use oxidative stress for wound healing, decontamination, immunomodulation of various skin pathologies, cancers, and respiratory viral infections, such as SARS-CoV-2 (Figure 2). The redox signaling molecules described in this review can modulate the gene and protein expression and affect the intracellular redox levels and cellular integrity. Therefore, it is paramount to understand the molecular response of endogenous antioxidants to the extracellular oxidant interaction. This review elaborates how oxidative stress-based therapeutic approaches can offer a promising way to prevent and treat human diseases.

## Figures and Tables

**Figure 1 biomedicines-10-00823-f001:**
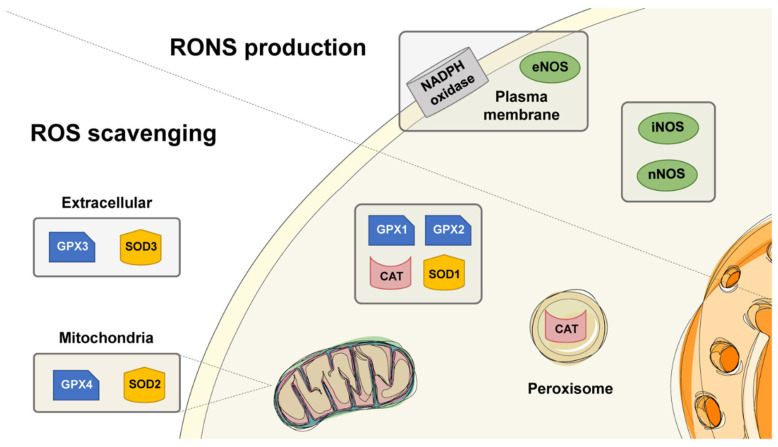
Cellular localization of the enzymatic systems related to oxidative stress.

**Figure 2 biomedicines-10-00823-f002:**
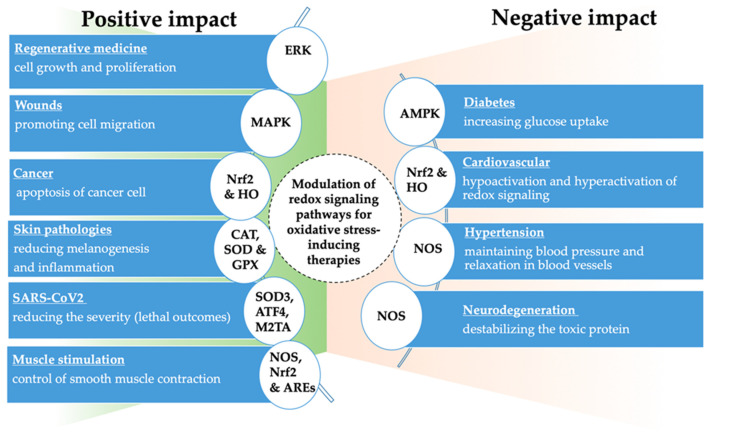
Medical conditions and summary of the known effects of oxidative stress-inducing therapies.

**Table 1 biomedicines-10-00823-t001:** List of oxidants that are relevant in biology, as well as the reactions and cellular effects.

Oxidants		Half-Life Time (s)	Cellular Source	Reaction
Name	Symbol			
Nitric oxide	NO•	<1	Nitric oxide syntase (NOS) enzyme	2 L-arginine + 3 NADPH + 3 H^+^ + 4 O_2_ ⇌ 2 citrulline + 2 NO• + 4 H_2_O + 3 NADP^+^
Superoxide anions	O_2_^−^	10^−6^	Mitochondrial electron transport chain, cell organelles	NADPH → NADP^+^ + H^+^ + 2e^−^ 2e^−^ + 2 O_2_ → 2 O_2_^−^
Hydrogen peroxide	H_2_O_2_	10^−5^	NOXs and mitochondrial respiratory chain	SOD 2 O_2_^−^+ 2H^+^ → H_2_O_2_ + O_2_
Hydroxyl radical	OH•	10^−9^	Fenton reaction as a result of interactions between H_2_O_2_ and metal ions	Me^n+^ + H_2_O_2_ → Me^(n+1)+^ + 2 OH• (Me represents a transition metal such as Fe, Mn, Cu, or Co)
Peroxynitrite	ONOO^−^	1	Reaction between O_2_^•^^−^ or O_2_ with NO• formed by iNOS	O_2_^−^ + NO• → ONOO^−^

**Table 2 biomedicines-10-00823-t002:** Reactions catalyzed by various intracellular antioxidant enzymes.

Intracellular Antioxidant Enzyme	Cellular Location	Oxidant	Concentration	Reaction Catalyzed
Superoxide dismutase (SOD)	SOD1: Cytoplasm SOD2: Mitochondria SOD3: Extracellular	O_2_^−^	Normal: 4–10 μM	O_2_^−^ → H_2_O_2_
Catalase (CAT)	Cytoplasm, peroxisome	H_2_O_2_	Plasma: in 1 nM Human blood cells: 2–3 μM	2 H_2_O_2_ → O_2_ + 2 H_2_O
Glutathione peroxidase (GPXs)	GPX1 and GPX2: Cytoplasm GPX3: Extracelllular GPX4: Mitochondria	H_2_O_2_	0.2 µm in red blood cells to values of 2.5 µm and 6.7 µm derived from mathematical models	H_2_O_2_ + 2 GSH → GSSG + 2H_2_O

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
