# Peer review of "Modulating the Antioxidant Response for Better Oxidative Stress-Inducing Therapies: How to Take Advantage of Two Sides of the Same Medal?"

_biomedicines, 2022, doi:10.3390/biomedicines10040823_

Round 1
Reviewer 1 Report
General aspects.
The manuscript of Shaw and co-workers is a complete and extensive review regarding the issue of redox balance and the relationship with pathology. At first sight, the focus of the manuscript seems to be the use of oxidative-stress inducing therapies (as the title states). This is a relevant aspect, much less explored, since oxidative stress is generally viewed as the culprit of many diseases. However, along the manuscript, the reader also finds many aspects of the pathological side of oxidative stress and the importance of antioxidants.
Main points
I think the manuscript has the value to summarize the different sides of the coin of oxidative stress. Therefore, I think this aspect needs to be explicitly stated in the manuscript, in the introduction and summary. Accordingly, the title could be changed in order to provide the view of redox balance from the two points of view, not only to oxidative stress inducing therapies.
I propose to organize the text in the following sequence (see comments for each section).
- Introduction. I suggest including in this section a summary of the organization of the manuscript, to set the question of the two sides of oxidative stress in health and disease. I also suggest being more broad in this section; for example, I would delete lines 7-9 of the second page “For example, quenching …”., since they are too specific for an introduction.
- Biologically relevant oxidants. Table 1 is adequate, but I would eliminate the last column about the cellular effect, since only some effects are shown among the many (beneficial and harmful) related to each oxidant. Regarding the aspect of NO in platelet aggregation, it should be noted that NO inhibits platelet adhesion and aggregation (from the text, it seems the opposite). On page 4, the last paragraph regarding the interaction between NO and extracellular superoxide anion, can you indicate the sources of the superoxide anion? NADPH oxidases? Xanthine oxidases?
- Enzymatic systems related to oxidative stress. Table 2 is adequate. When referring to the different SODs, it would be worth mentioning their locations in the cell or outside the cell. Similarly, please refer to catalase location.
- Redox signaling networks. I think this is the less clear section. It is not obvious from the text why these particular pathways were chosen: because oxidative stress has an influence on them? or because they are intermediate pathways between oxidative stress and disease states. This aspect should be clarified. Table 3 is a bit confusing too.
- Applications of oxidative stress in health care.
- Source of oxidants for treatment. In the subsection about laser Therapy, Ref 111 refers to male infertility. This part of the section is focused on tissue regeneration and use in cancer, since you included this reference, perhaps it is worth mentioning this aspect in the text.
Figure 1 does not summarize well the manuscript, neither reflects the complexity of the issue about oxidative stress in health and disease. I think you tried to recap too much in one figure. The manuscript would benefit from more than one schematic diagrams in some of the sections, rather than a single one.
References. Some of references have more than 20 years. If they are not essential for the understanding, perhaps you could consider deleting them, since the review is overall sufficiently referenced.
Reference 8. Missing information
Ref 58 refers to plant biology. Unless it is essential, I suggest to eliminate since the manuscript is centered in human physiology.
Author Response
Reviewer 1
We thank the reviewer for the time and efforts made to examine our paper. We are happy that the reviewer felt that our work is interesting and are grateful for the suggestions to improve the manuscript and make it more complete. Please see our answers to the comments below (in blue) regarding the steps we have taken to address the issues raised in the reviewer’ reports. The corresponding changes are highlighted in yellow below and in the revised manuscript.
General aspects.
The manuscript of Shaw and co-workers is a complete and extensive review regarding the issue of redox balance and the relationship with pathology. At first sight, the focus of the manuscript seems to be the use of oxidative-stress inducing therapies (as the title states). This is a relevant aspect, much less explored, since oxidative stress is generally viewed as the culprit of many diseases. However, along the manuscript, the reader also finds many aspects of the pathological side of oxidative stress and the importance of antioxidants.
We thank the reviewer for the positive comment.
Main points
I think the manuscript has the value to summarize the different sides of the coin of oxidative stress. Therefore, I think this aspect needs to be explicitly stated in the manuscript, in the introduction and summary. Accordingly, the title could be changed in order to provide the view of redox balance from the two points of view, not only to oxidative stress inducing therapies.
We have changed the title of the review paper as “Modulation of antioxidant response for better oxidative stress-inducing therapy: how to take advantage of two sides of the same medal?”
I propose to organize the text in the following sequence (see comments for each section).
We thank the reviewer for this suggestion, we have reorganized the sections.
- Introduction. I suggest including in this section a summary of the organization of the manuscript, to set the question of the two sides of oxidative stress in health and disease. I also suggest being more broad in this section; for example, I would delete lines 7-9 of the second page “For example, quenching …”., since they are too specific for an introduction.
We have modified the text as follows:
Section 1, Page 2, line 55:
It is essential to recognize how these antioxidant defense systems can be targeted for therapeutic use, considering the two faces of oxidative stress in health and disease. This review article presents the biologically relevant oxidants and their chemistry, the enzymatic systems and the redox signaling networks involved in the response to oxidative stress, the therapeutic use of oxidative stress in health care and the current external sources of reactive species for disease treatment. We discuss the relationship of pro-oxidant therapies with the antioxidant mechanisms in disease progression, including ulcers, skin pathologies, oncology, and viral infections such as SARS-CoV-2.
Also, we have removed the sentence (line 7-9 of the second page) from the introduction, and modified the text as follows, to make it less detailed:
Section 1, Page 1, line 41:
In the same way, therapies that generate high levels of RONS could be used to treat a variety of diseases, including neurodegenerative disorders [5], respiratory diseases [6], various types of cancers [7-10], among others.
- Biologically relevant oxidants. Table 1 is adequate, but I would eliminate the last column about the cellular effect, since only some effects are shown among the many (beneficial and harmful) related to each oxidant.
We have removed the last column from Table 1.
- Regarding the aspect of NO in platelet aggregation, it should be noted that NO inhibits platelet adhesion and aggregation (from the text, it seems the opposite).
We thank the reviewer for pointing this out. We have corrected the text:
Section 2.1, page 3, line 90:
“Furthermore, NO• suppresses platelet aggregation…”
- On page 4, the last paragraph regarding the interaction between NO and extracellular superoxide anion, can you indicate the sources of the superoxide anion? NADPH oxidases? Xanthine oxidases?
We have added the following information:
Section 2.2, page 4, line 132:
In recent studies, it has been found that extracellular O2- produced by LPS-stimulated macrophages induced Ca2+ mediated signaling and cell death in pulmonary endothelial cells [39].
- Enzymatic systems related to oxidative stress. Table 2 is adequate. When referring to the different SODs, it would be worth mentioning their locations in the cell or outside the cell. Similarly, please refer to catalase location.
We have added this information to the text and to Table 3 (cellular location).
Section 3.3, Page 7, line 249:
In humans, SOD binds to three metals: Cu/Zn (SOD1, cytosol), manganese (SOD2, mitochondria), and Cu/Zn (SOD3, extracellular) [81]. Cellular SOD concentrations are…
- Redox signaling networks. I think this is the less clear section. It is not obvious from the text why these particular pathways were chosen: because oxidative stress has an influence on them? or because they are intermediate pathways between oxidative stress and disease states. This aspect should be clarified.
We have modified the text in Section 4 to better reflect why these pathways were chosen.
Section 4, Page 8, line 317:
Oxidative stress disturbs cellular ROS homeostasis via a redox relay mechanism [94], which leads to an increase in intracellular ROS levels and modulates key cell signaling pathways. Redox signaling is also crucial in regulating tumorigenesis, autoimmunity, neurodegenerative diseases, and loss of tissue regeneration with age [79,95,96]. Here we describe some of the key networks activated upon oxidative stress, involved in health and disease.
Section 4.2, Page 9, line 335:
4.2 The Keap1-Nrf2-ARE pathway
Kelch-like ECH-associated protein 1 (Keap1) is referred to as a negative regulator of the nuclear factor E2-related factor (Nrf2) [101]. It participates in cellular defense against various exogenous and endogenous stressors [59,101] and hence, it is a potential target for many drugs for the control of diseases. NRF2 belongs to the cap'-n'-collar subfamily of the basic-region leucine zipper bZIP transcription factors and binds to the cis element electrophile response element (EpRE) and other antioxidant response elements (AREs) in DNA. Together, they regulate the expression of more than 200 genes involved in antioxidant defence, DNA repair, proteasome activity, among other processes required for cell survival [14]. The system is induced by electrophiles such as H2O2 and other intermediary metabolites. In addition, Nrf2 inducers such as itaconate and tert-butylhydroquinone react with Keap1 (cysteine thiol groups), resulting in a defensive antioxidant response through the activation of the Keap1-Nrf2 signaling pathway (Figure 2) [102,103]. Upon oxidative stress, Nrf2 is released, eludes degrada-tion, and translocates to the nucleus, where it starts its transcriptional activity on target genes. The dissociation of Nrf2 is mainly due to the conformational change in oxidized Keap1 (disulfide bond formation between Cys273 and Cys288). Multiple factors such as musculoaponeurotic fibrosarcoma and AREs in the Keap1-Nrf2 signaling pathway are responsible for the protection of cells under oxidative stress [104,105]. Any mutation in the Keap1 promoter or modification protein reduces its expression, resulting in a higher expression of Nrf2, which consequently favors cancer cell survival [106]. So, activation of Nrf2 might be a key target for cancer therapy [107,108]. Apart from cancer therapy, Nrf2 can also be a promising target for the development of neuroprotective and antidiabetic drugs [109,110].
In addition, we have added a section on the NF-κB pathway, that is relevant in the response to oxidative stress:
Section 4.4, Page 9, line 369:
4.4 NF-κB pathway
The transcription factor nuclear factor κB (NF-κB) is localized in the cytoplasm as a heterodimer. Once activated, NF-κB translocates into the nucleus to induce gene transcription of genes related to cell survival. The activation of NF-κB occurs at the early stages of oxidative stress, but it is reduced under sustained presence of ROS [114]. ROS activates the canonical pathway by inhibiting the phosphorylation of IκBa (required to block the binding of NF-κB to the DNA) and inhibiting IKKβ activity [54].
The NF-κB activity influences the levels of ROS in the cell by increasing the expression of antioxidant proteins such as CAT, SOD, GPX, ferritin heavy chain, thioredoxins and glutathione S-transferase pi, HO-1, among others. Conversely, ROS can repress NF-κB signaling, depending on the phase and context. The direct oxidation of NF-κB on the cysteine residue Cys-62 in the RHD domain of the p50 subunit inhibits NF-κB binding to the DNA [115]. In addition, the NF-κB pathway regulates the expression of pro-oxidant targets such as iNOS [116], cytochrome p450 enzymes [117], xanthine oxidase/dehydrogenase [115,117,118], NADPH oxidase NOX2 subunit gp91phox [114], among others.
- Table 3 is a bit confusing too.
We thank the reviewer for this comment. We believe the reviewer refers to Table 4, regarding to redox signaling networks. As most of the information is presented in the main text, we have removed this table from the manuscript.
- Applications of oxidative stress in health care.
- Source of oxidants for treatment. In the subsection about laser Therapy, Ref 111 refers to male infertility. This part of the section is focused on tissue regeneration and use in cancer, since you included this reference, perhaps it is worth mentioning this aspect in the text.
We thank the reviewer for this observation. However, we opted for removing the reference and do not add more information, since the manuscript is already quite extensive.
Figure 1 does not summarize well the manuscript, neither reflects the complexity of the issue about oxidative stress in health and disease. I think you tried to recap too much in one figure. The manuscript would benefit from more than one schematic diagrams in some of the sections, rather than a single one.
We thank the reviewer for this comment. We have modified the original figure and added a new one for clarity:
Section 3, Page 6, line 211:
Figure 1. Cellular localization of the enzymatic systems related to oxidative stress.
Section 5, Page 10, line 385:
Figure 2: Medical conditions and summary of the known effects of oxidative stress-inducing therapies

References. Some of references have more than 20 years. If they are not essential for the understanding, perhaps you could consider deleting them, since the review is overall sufficiently referenced.
Reference 8. Missing information
Ref 58 refers to plant biology. Unless it is essential, I suggest to eliminate since the manuscript is centered in human physiology.
We have revised the reference list and corrected it accordingly to keep the most relevant and up-to-date references.

Reviewer 2 Report
Submitted review summarizes available data regarding the potential of therapeutic application of induced oxidative stress. Different reactive species and antioxidants are described, sometimes too detailed, and their overall effect is analyzed in various diseases, including COVID-19.
Certain corrections, which are listed, are needed and necessary in order to make this article acceptable:
-part 1. Introduction: replace „keepers of oxidative balance“ with a more suitable term; add comma (,) in the last sentence of the section, after „oxidants“
- try to reorganize the text so that Table 1 is not split on 2 pages
- correct the name of NOS in Table 1
- rephrase cellular effects of H2O2 in Table 1
- section 2.1. – „in an out to the“ replace with through or rephrase the whole sentence
- make sure subscripts and superscripts are appropriately used (Ca2+)
- section 2.3. should be shorter and H2O2 is not a reactive species! Make sure to correct first sentence
- Table 2 should move closer to pages 6 or 7, when it is mentioned for the first time
- use more appropriate term for „levels of MAPK“ signaling and rephrase the following sentence
- section 3.1.1. – rephrase sentences 2 and 3 and thoroughly check language and grammar throughout the section; rephrase „non-malignancy conditions“
- correct „synergy on“ into „synergy with“
- section 3.3. change subtitle; Why are NOS and SOD, GPX and CAT grouped together?! Please correct
- 3.3.1. check language, eg. „hepatocytes enzymes“
- 3.3.2. In addition, they also......ref 136 – rephrase
- 3.3.4. replace „scout“ with more appropriate term; replace „can cause“ with „ is associated with“ various diseases
- 3.3.5. remove „a subcellular compartments such as“; mention extracellular localization and function of GPX3
- 4.1. check language „in vivo studies indicate MAPK is involved“; What do you mean by: ....produce insulin by utilizing excess blood glucose?
- 4.2. „stressors“ instead „stressess“; reference for the last sentence is lacking
- 4.3. reference for the last sentence is lacking
- 5.2. the first sentence is to colloquial, please rephrase
- you mention apoptotic proteins, specify which: pro- or anti-apoptotic; which protein is inactivated?
-5.3. replace „this evidences“, verb is used incorrectly
- 5.5.rephrase subtitle; replace „more experimentation“ with adequate term
Author Response
Reviewer 2
We thank the reviewer for the time and efforts made to examine our paper. We are happy that the reviewer felt that our work is interesting and are grateful for the suggestions to improve the manuscript and make it more complete. Please see our answers to the comments below (in blue) regarding the steps we have taken to address the issues raised in the reviewer’ reports. The corresponding changes are highlighted in yellow below and in the revised manuscript.
Submitted review summarizes available data regarding the potential of therapeutic application of induced oxidative stress. Different reactive species and antioxidants are described, sometimes too detailed, and their overall effect is analyzed in various diseases, including COVID-19.
We thank the reviewer for the positive comments.
Certain corrections, which are listed, are needed and necessary in order to make this article acceptable:
-part 1. Introduction: replace „keepers of oxidative balance“ with a more suitable term; add comma (,) in the last sentence of the section, after „oxidants“
The text has been modified as follows:
Section 1, Page 2, line 52:
The main therapeutic benefits of pro-oxidant treatments can be expected from the generation of oxidants that cause (i) inhibition of endogenous antioxidants or (ii) activation of cellular activity.
It is essential to recognize how these antioxidant defense systems can be targeted for therapeutic use, considering the two faces of oxidative stress in health and disease.
Section 2, Page 3, line 81:
These all are important factors required to elucidate the physiological actions of oxidants and to better understand how oxidants transduce their cellular signals, as well as how they regulate the antioxidant response.
- try to reorganize the text so that Table 1 is not split on 2 pages
- correct the name of NOS in Table 1
- rephrase cellular effects of H2O2 in Table 1
- section 2.1. – „in an out to the“ replace with through or rephrase the whole sentence
- make sure subscripts and superscripts are appropriately used (Ca2+)
We have modified and arranged the text in and around Table 1 so that Table 1 is in one page. Also, we have corrected the text according to these comments.
- section 2.3. should be shorter and H2O2 is not a reactive species! Make sure to correct first sentence
We thank the reviewer for the suggestion. We have shortened the section as follows:
Section 2.3, Page 4, line 139:
2.3 Hydrogen peroxide (H2O2)
H2O2 is the least reactive oxygen species and remains stable under physiological pH and temperature in the absence of metal ions in vivo. The intracellular production of H2O2 has mainly a regulatory role and the normal intracellular steady-state concentration of H2O2 is 10 nM or below [40]. The majority of H2O2 produced by mitochondria is initially originating from O2−, which is produced by mitochondrial enzyme complexes [41,42]. The H2O2 generation from mitochondria is in the range of 50 µmol kg-1 min-1 [43].
H2O2 is continuously produced in vivo [44] at physiological levels. H2O2 is a major component in redox signaling [44,45] and the major redox species operative in redox sensing, signaling, and redox regulation [46,47]. For such a function, a controlled transport of H2O2 across membranes is required. It has been shown that the permeability of the plasma membrane to H2O2 is a significant factor in cell susceptibility to extracellular H2O2 [48]. Extracellular H2O2 can diffuse into the cell, where it is rapidly decomposed by the intracellular antioxidants. However, the spatial distribution of H2O2 in cells and tissues is not uniform. There are substantial gradients resulting from the different concentrations of antioxidant enzymes in the cell compartments [44].
In addition, the integral membrane proteins that transport water, aquaporins (AQP), can also transport H2O2 within the cell [49], and they have been suggested as biomarkers for disease. AQP expression during stress conditions regulates the cell membrane integrity as well as cell communication [49]. AQP3 expression in human pancreatic cancer cell lines enhanced intracellular H2O2 levels in vitro (H2O2 = 80-90 µM) [48], highlighting its role in controlling the influx of H2O2 through the plasma membrane. In wound healing, H2O2 signaling has been established [48,50]. However, H2O2 may induce lipid peroxidation [7,51] at concentrations > 150 µM [52]. Furthermore, exogenous H2O2 can cause phosphorylation of tyrosine and activation of growth factors [53,54]. The pathological relevance of extracellular H2O2 is also well recognized in inflammation and injury responses [55].
Regarding H2O2, it is indeed a non-radical oxidant, hence less reactive than the other oxidants, but it is still called a reactive oxygen species, as described in the literature (e.g., Gough et al, 2011). For this reason, we have not changed the text.
Gough, D.R.; Cotter, T.G. Hydrogen peroxide: a Jekyll and Hyde signalling molecule. Cell Death Dis 2011, 2, e213, doi:10.1038/cddis.2011.96.
- Table 2 should move closer to pages 6 or 7, when it is mentioned for the first time
We have moved it immediately after the text where it is first mentioned.
- use more appropriate term for „levels of MAPK“ signaling and rephrase the following sentence
We have modified the text as follows:
Section 6, Page 15, line 650:
There is a connection between the levels of oxidants in a cell and the activation of MAPK signaling.
- section 3.1.1. – rephrase sentences 2 and 3 and thoroughly check language and grammar throughout the section; rephrase „non-malignancy conditions“
We have modified the section as suggested:
Section 6.1.1 Radiotherapy, Page 15, line 666
6.1.1 Radiotherapy
Radiation therapy uses high-energy particles or waves, such as x-rays, gamma rays, electron beams, or protons that generate RONS [209,210] (Table 3), and induce intracellular oxidative stress in the subcellular compartments [211,212]. The standard therapy involves the application of a total dose of 40-50 Gy for the treatment of breast cancer [213] and rectal cancer [214]. However, the recommended doses sometimes fail to ablate the tumor and can lead to resistance due to inherent characteristics of the tumor [215]. In addition, the presence of cancer stem cells (CSCs) in the tumor can contribute to the resistance: CSCs have low intracellular ROS levels, an increased expression of ROS scavengers, an efficient DNA repair system, and can inhibit apoptosis [216].
Besides cancer therapy, low doses of radiotherapy can be used to treat non-neoplastic degenerative, chronic inflammatory or proliferative diseases, as it blocks different inflammatory mediators and promotes the production of anti-inflammatory cytokines [217]. For example, the treatment with < 1 Gy can inhibit iNOS expression, and reduce endothelial cell-leukocyte interactions due to a decrease in the expression of adhesion molecules, and reduced vasodilatation [218].
- correct „synergy on“ into „synergy with“
We have made the correction.
- section 3.3. change subtitle; Why are NOS and SOD, GPX and CAT grouped together?! Please correct
We have changed it to better reflect the content of the section:
Section 3. Page 5, line 202:
Enzymatic systems related to oxidative stress
- 3.3.1. check language, eg. „hepatocytes enzymes“
We have corrected the sentence:
Section 3.1, Page 6, line 224:
Remarkably, it has been shown that during the liver infection, a certain number of hepatic enzymes, including iNOS, localize to peroxisomes
- 3.3.2. In addition, they also......ref 136 – rephrase
We have corrected the text as follows:
Section 3.2, Page 6, line 238:
The regulation and function of each NOX remain unclear, but it is believed that they are key mediators of normal biological responses. In addition, NOXs contribute to various diseases such as neuron inflammation, neurodegeneration, cardiovascular, renal disease, including hypertension, and atherosclerosis [79].
- 3.3.4. replace „scout“ with more appropriate term; replace „can cause“ with „ is associated with“ various diseases
The following sentences have been modified:
Section 3.4, Page 7, line 271:
It is present in practically all types of living cells, where it scavenges H2O2 molecules.
Section 3.4, Page 7, line 276:
The mutation or deficiency of the CAT enzyme is associated with various diseases and abnormalities in humans.
- 3.3.5. remove „a subcellular compartments such as“; mention extracellular localization and function of GPX3
The localization and function of GPXs has been added to the text, as well as to Table 2.
Section 3.5, Page 8, line 289:
Glutathione Peroxidase (GPX) is a selenium-dependent enzyme present in the cytosol (GPX1 and GPX2), mitochondria (GPX4), and extracellular space (GPX3) [81,88].
Section 3.5, Page 8, line 302:
Under normal conditions, it has been proposed that the extracellular GPX3 could protect cells from oxidative damage [89]. However, GPX3 has been involved in cancer, as it can suppress tumor progression in cancer cells exposed to oxidative stress [90].
- 4.1. check language „in vivo studies indicate MAPK is involved“; What do you mean by: ....produce insulin by utilizing excess blood glucose?
We have removed the sentence for clarity.
- 4.2. „stressors“ instead „stressess“; reference for the last sentence is lacking
- 4.3. reference for the last sentence is lacking
We have corrected the text and added the following references to the manuscript:
Section 4.2, Page 9, line 356:
Apart from cancer therapy, Nrf2 can also be a promising target for the development of neuroprotective and antidiabetic drugs [109,110].
References:
- Zhao, J.P.; Liu, L.R.; Li, X.; Zhang, L.X.; Lv, J.; Guo, X.L.; Chen, H.; Zhao, T.F. Neuroprotective effects of an Nrf2 agonist on high glucose-induced damage in HT22 cells. Biol Res 2019, 52, doi:10.1186/s40659-019-0258-z.
- David, J.A.; Rifkin, W.J.; Rabbani, P.S.; Ceradini, D.J. The Nrf2/Keap1/ARE Pathway and Oxidative Stress as a Therapeutic Target in Type II Diabetes Mellitus. J Diabetes Res 2017, 2017, doi:10.1155/2017/4826724.
- 5.2. the first sentence is to colloquial, please rephrase
We have modified the sentence as follows:
Section 5.2, Page 11, line 435
In the skin, multiple biochemical processes take place, including the generation of ROS (Figure 2).
- you mention apoptotic proteins, specify which: pro- or anti-apoptotic; which protein is inactivated?
We have modified the text as follows:
Section 5.2, Page 11, line 449:
It has been found that melanocytes from patients with active vitiligo are exposed to abnormal levels of oxidative stress, which could be a consequence of abnormal mitochondrial ROS production, low production of antioxidants, and expression of pro-apoptotic proteins such as TRAIL [137,138].
References:
- Passi, S.; Grandinetti, M.; Maggio, F.; Stancato, A.; De Luca, C. Epidermal oxidative stress in vitiligo. Pigment cell research 1998, 11, 81-85, doi:10.1111/j.1600-0749.1998.tb00714.x.
- Zhu, L.F.; Lin, X.; Zhi, L.; Fang, Y.S.; Lin, K.M.; Li, K.; Wu, L.C. Mesenchymal stem cells promote human melanocytes proliferation and resistance to apoptosis through PTEN pathway in vitiligo. Stem Cell Res Ther 2020, 11, doi:10.1186/s13287-019-1543-z.
-5.3. replace „this evidences“, verb is used incorrectly
Thank you. The word has been replaced by “suggests” (Page 13, line 519).
- 5.5.rephrase subtitle; replace „more experimentation“ with adequate term
We have modified the text as follows:
Section 5.5, Page 15, line 631:
To date, even though data have been collected, the efficacy of such treatments targeting oxidative stress is still controversial and more research is necessary
